# Increasing Prevention of Mother to Child Transmission (PMTCT) Uptake through Facility-Based Health Promotion: Intervention Development

**DOI:** 10.3390/bs13040317

**Published:** 2023-04-06

**Authors:** Ibrahim Elsiddig Elsheikh, Rik Crutzen, Ishag Adam, Salah Ibrahim Abdelraheem, Hubertus W. Van den Borne

**Affiliations:** 1Department of Health Promotion, CAPHRI School for Public Health and Primary Care, Maastricht University, 6200 MD Maastricht, The Netherlands; 2Sudanese Public Health Association (SPHA), Khartoum 11111, Sudan; 3Department of Obstetrics and Gynecology, Unaizah College of Medicine and Medical Sciences, Qassim University, Unaizah 56219, Saudi Arabia; 4Department of Obstetrics and Gynecology, Omdurman Maternity Hospital, Khartoum 24983, Sudan

**Keywords:** PMTCT, pregnant women, intervention mapping, health promotion

## Abstract

In Sudan, the HIV testing rates during pregnancy remain low. Limitations in scaling and uptake of PMTCT services are linked to several factors within the healthcare system, including the motivation of healthcare providers. This article describes how a health facility-based health promotion intervention plan was developed, implemented, and evaluated to increase the uptake of PMTCT services using the Intervention Mapping approach. Individual-level and environmental determinants were previously identified and included in the intervention plan. Some factors that impacted the intention of women to test for HIV during pregnancy include level of knowledge on MTCT, who offers the HIV test, the fear and tension experienced when thinking about HIV/AIDS, the non-confidentiality of the HIV test results, and self-efficacy. This provides insights into how to develop, implement, and evaluate a facility-based health promotion intervention. The pre-assessment was critical in shaping the intervention and making it relevant and evidence based. The Intervention Mapping approach that was applied facilitated the systematic design of the intervention and supported guiding the implementation.

## 1. Introduction

It is estimated that 38.4 million people globally were living with HIV in 2021 and 1.5 million became newly infected [1]. More than 1500 children are infected with HIV every day around the world and more than 90% of such children were infected through mother-to-child transmission (MTCT) [2,3]. In sub-Saharan Africa, women and girls accounted for 63% of all new HIV infections in 2021 [1]. Previous studies revealed that the sub-Saharan African region is the most affected globally, recording the highest rates of new HIV infections [4]. Furthermore, 59% of new HIV infection cases driven by heterosexual transmissions are among women of reproductive age. This emphasizes the importance of implementing effective strategies to reduce MTCT to achieve zero new HIV infections among children. Prevention of Mother to Child Transmission (PMTCT) services can reduce the risk of MTCT effectively [3]. It includes services such as testing for HIV during pregnancy or among women of childbearing age, enrolling in HIV treatments as early as the second trimester [5], labour management (Caesarean section), and making information on breastfeeding and paediatric care available to promote informed decisions [6,7]. In Sudan, PMTCT services began in 2005 in a few states as a pilot, and by 2013, it scaled up to 227 sites. Provider Initiated Testing and Counselling (PITC) was introduced in 2012 [8]. Although PMTCT services have been made available through primary, secondary, and tertiary levels of healthcare, the complete implementation of PMTCT packages is available only at tertiary levels. Even though PMTCT services can effectively reduce the risk of MTCT and are available free of charge (including HIV testing), access to antenatal HIV testing and antiretroviral (ARV) therapy are as low as <1% and <10%, respectively [6]. In Sudan, the PMTCT uptake is low, and the reluctance in uptaking PMTCT services can be traced to several factors associated with healthcare such as the hesitancy from the healthcare workers to provide HIV testing to pregnant women, lack of accountability, limited training of healthcare providers, and the failure to inculcate HIV testing as a part of the routine check-ups for pregnant women [8]. The incomplete adoption of the opt-out strategy regarding HIV testing, which was piloted in Sudan in 2007 and implemented in 2009, is also attributed to be a major drawback in the scaling-up of PMTCT services [6,9]. 

In addition to the health system and health facility-related issues, social norms, taboos, and acceptance play an impactful role as barriers to PMTCT service uptake [10], which can be alleviated with counselling and supportive healthcare workers. The low level in PMTCT uptake is also associated with behavioural factors such as stigma, discrimination, poor staff–client relationships, lack of accessibility to services, and insufficient knowledge about HIV among pregnant women. Long waiting times, unfriendly staff attitude, high transport costs, and low service accessibility due to fragmented delivery have been noted as key factors that affect PMTCT and enrolment of women in ARV [11]. A study in Zimbabwe showed that a mother-to-mother peer support group was an effective method to raise awareness and retain women for PMTCT programs [12]. Individual factors such as financial situation and perceptions of stigma are limiting factors for PMTCT uptake [13]. Only a few studies have been conducted in Sudan about maternal HIV infection [14] and about identifying the determinants and attitudes of pregnant women towards voluntary counselling and HIV testing (VCT) services [6]. In a study conducted at Khartoum Hospital, Sudan, 1005 women were interviewed at the perinatal medicine section, which offers services free of charge. Though 55.9% (*n* = 562) of women had some idea about MTCT, only 30.3% (*n* = 305) were tested [14].

The limited insight into the causes of this problem in Sudan prompted us to study the perceptions of pregnant women in Sudan and also explore individual- and health facility-level related factors. Hence, we conducted qualitative and quantitative research to obtain insights into the perceptions of Sudanese pregnant women towards HIV infections and the available PMTCT services [15,16]. The findings of these studies were used to design an intervention and develop an implementation plan for this intervention. The aim of this article is to describe the intervention development. It provides details on how a health facility-based health promotion intervention/plan was developed, implemented, and how it will be evaluated based on the Intervention Mapping (IM) approach. 

## 2. Materials and Methods

The intervention plan was developed based on the Intervention Mapping approach [17]. Intervention Mapping (IM) is a systematic approach that is built mainly on three perspectives: (1) using theory and evidence as foundations to develop interventions for behaviour and environmental changes to avoid reinventing the wheel; (2) application of a social-ecological model that views the individual behaviour as an outcome of the interaction of the individual with their environment; and (3) community engagement and participation to ensure relevancy [17]. IM applies six key steps: (1) conducting a need assessment, (2) stating program outcomes and objectives, (3) designing the program, (4) producing the program, (5) planning program implementation, and (6) planning program evaluation [17,18]. In each step, the developers apply findings from theory, evidence, and their own research [18]. Ethical clearance was obtained from the Ministry of Health. Written informed consent was not feasible due to sensitivity issues around HIV/AIDS. Hence, verbal informed consent was obtained from all women who participated in this study. The below illustration in Figure 1 is adapted and based on IM. It describes our six major areas starting by having the need assessment, intervention design, and intervention assets and materials, followed by implementation, and, finally, evaluation. 

### 2.1. Step 1: Need Assessment

In order to understand the determinants associated with the intention and behaviour of HIV testing during pregnancy, we have carried out a formative assessment and a cross-sectional survey [15,16]. This helps in identifying individual-level determinants and environmental and contextual factors. Discussions were held with health workers and hospital management teams as part of the need assessment. 

### 2.2. Step 2: Program Outcomes and Objectives 

In this step, we define what needs to change in the behaviours of the priority population (i.e., pregnant women) as well as agents in the environment (e.g., PMTCT teams and hospital management). We made use of evidence collected in the assessment phase and designed program outcomes and change objectives for the priority population and environmental agents. 

### 2.3. Step 3: Program Design 

This step refers to selecting theory- and evidence-based methods and developing them into practical applications to support the program design. We identified methods that can influence the determinants, and we linked them to the change objectives identified in Step 2 above. Several discussions were held with the research team, PMTCT teams (physicians, counsellors, and lab technicians), and hospital management to discuss the pre-assessment and facility-based intervention plan. We focused on tailored interventions targeting three groups: the patients (pregnant women), the PMTCT team (counsellor, lab technician and doctor), and the hospital management. 

### 2.4. Step 4: Developing Intervention Components and Program Production 

The practical applications were arranged into creative program components and materials to support the achievements of the intervention objectives. In doing so, we ensured that the program messages and intervention components fit both the priority population and the environmental agents as well as the context in which it will be implemented. We have achieved that through consultation meetings with various teams and also via pre-testing of these products and materials among their intended users. 

### 2.5. Step 5: Implementation Plan 

An implementation plan that takes into consideration all previous steps was developed to support behavioural outcomes and address environmental and contextual factors. Change objectives were developed to address each determinant in relation to the PMTCT service uptake. The main objective of the intervention plan was to contribute to increase HIV testing during pregnancy, which is a core component in the PMTCT program. Our research was planned in four hospitals. Two hospitals as the experimental group and the other two as the control (stratified according to size; Table 1). There were similarities in patient flow, the functionality of the PMTCT, and the patient’s socioeconomic status among the selected hospitals. Omdurman and Al Turki maternity hospitals were selected for the experimental group, after agreement with the hospital management, and two comparable hospitals were identified for the control group. Additionally, the socioeconomic status of the catchment areas around these four hospitals has a lot of similarities. PMTCT sites were established in these hospitals within the ANC clinics where pregnant women come for routine ANC checkups. They were asked to voluntarily participate in this study. There were usually long waiting times so most of the women found it beneficial to participate in the group awareness and counselling sessions and also listen/watch the audio/visual materials. They were told that enrolment in PMTCT is free of charge. No financial incentives were given. Following the evaluation of this intervention/plan, recommendations will be made available to be implemented in all hospitals that offer PMTCT services. 

### 2.6. Step 6: Evaluation Plan

We developed an evaluation plan to determine the impact of our intervention. The assumption was that if the intervention plan is adequately implemented, we will see an increase in the number or percentage of women enrolled in PMTCT (first step HIV test) at the hospitals in the experimental group compared to those in the control group. We also expected to influence the environmental and behavioural outcomes through our intervention. We carried out qualitative and quantitative data and designed indicators to assess the effectiveness of the health facility-based health promotion intervention in increasing the uptake of PMTCT services.

## 3. Results 

### 3.1. Step 1: Need Assessment

This section provides an overview on the key determinants related to PMTCT and HIV testing during pregnancy among pregnant women in Sudan. It also covers the context and health facility factors in relation to implementation of the PMTCT services. There are many factors that impact the uptake of PMTCT services during pregnancy [11]. To better understand this in the Sudan context, we carried out a formative assessment and a cross-sectional survey [15,16]. 

The need to have sufficient knowledge about MTCT: The international literature showed that comprehensive knowledge about MTCT helps pregnant women to decide about PMTCT services [19,20]. In our assessment, we found that most participants were aware of the sexual transmission of HIV [15,16], but few knew about MTCT of HIV. None mentioned MTCT during child labour/delivery [15]. 

Advantages of HIV test during pregnancy: In our assessment, many participants failed to see any advantage of HIV testing during pregnancy and mentioned stigma, fear, and mistrust of result confidentiality as the major disadvantages. Several women perceived HIV as a burden and unnecessary, failing to recognize the risks of MTCT [15]. On the other hand, some participants stated that HIV testing was vital as it can help with medications to save the child and making informed decisions when it comes to labour/delivery and breastfeeding [15]. Those who believed HIV testing during pregnancy had advantages still associated HIV testing with fear, stigma, and suspicion leading to not testing themselves [16]. 

The need to make an informed decision about HIV test during pregnancy: The pregnant women need to know what it takes before they enrol in PMTCT. The first step to enrol in the PMTCT services is to take an HIV test. In our assessment, the pregnant women showed fear and tension when they thought about the HIV test. This is also linked to anticipated stigma and dealing with consequences that they are not prepared for [15,16]. 

The need for assistance when making decision: Pregnant women need some support when they decide to enrol in the PMTCT. The PMTCT protocol provides pre- and post-HIV counselling for those who want to undergo HIV testing during pregnancy and follow-up care services for those who test positive [9]. In our assessment, the pregnant women showed hesitancy related to the confidentiality of the test results and limited family support [15,16]. 

The need to have approval to conduct HIV test during pregnancy: In our assessment, for the pregnant women, doctors were found to play the most influential role in the decision to receive or refuse HIV testing, followed by their husbands and their mother. Husbands did not have a significant role [16]. Most women indicated that they had never been tested before but were showing a willingness to be tested during pregnancy. For those who had tested before, most of them stated that the HIV test was offered to them by doctors, only a very few requested it themselves. Younger women were most likely to accept HIV tests during pregnancy [16]. 

Perceived susceptibility and self-efficacy: The international literature showed that sociocognitive determinants (e.g., self-efficacy, perceived susceptibility) had a significant relation with HIV testing intention [21,22]. In our assessment, we found that most of the participants had high perceived severity, high susceptibility, and high self-efficacy with regard to HIV. Almost half of the participants had a positive attitude toward HIV testing. Women with higher self-efficacy and higher perceived susceptibility were more likely to have the intention to be tested [16]. 

The summary of the need assessment identified key factors that impact the intention of women to test for HIV during pregnancy. These are (1) the level of knowledge about MTCT, (2) the profession of the person offering the HIV test, (3) the fear and tension experienced when thinking about AIDS/HIV, (4) the perceptions of the pregnant women on the non-confidentiality of the HIV test results, and (5) self-efficacy [15,16]. These are in addition to the environmental factors identified: for example, buy-in from the hospital management, availability of testing kits, and commitment from the laboratory and counselling staff. Hence, a comprehensive package of health promotion interventions targeting pregnant women and the environment where PMTCT works was required. Our intervention components focused on addressing the above factors.

### 3.2. Steps 2 and 3: Program Outcomes, Objectives and Design 

Following our assessment, we designed a logic model for this intervention, and we selected change methods that were translated into practical applications. A technical advisory group was formed, and a budget for the implementation was prepared and successfully secured. The Ministry of Health and the hospitals provided approval and ethical clearance to conduct this intervention.

One of the key strategies used for the PMTCT services is an opt in/out policy where the pregnant women are given the chance to opt in or opt out, meaning the HIV test during pregnancy is voluntary [6,9]. In our assessment, we found that many women were not opting in [16]. Individual level and environmental determinants need to be addressed in order to encourage pregnant women to opt in and enrol in the PMTCT [15,16]. The environmental changes were doctors offering the HIV test for pregnant women while they attend their follow-up ANC visits, additional counselling as required provided by counsellors, lab technicians on the ready to perform increased HIV test requests, and securing HIV testing kits throughout the intervention period (Table 2). The behavioural outcome expected was pregnant women attending the ANC clinics and accepting HIV tests. On the individual level determinants, the pregnant women need to be educated, encouraged, and counselled to take HIV tests (Table 3). 

### 3.3. Steps 4 and 5: Developing Intervention Components and Implementation Plan 

Based on the previous three steps, we have produced all components of the intervention. The intervention plan, which was based on the need assessment, was discussed with the hospital management and PMTCT teams. Following the approval of the plan, the interventions were executed at Omdurman Maternity hospital and Al Turki Maternity hospital for a period of six months. The other two hospitals were left as a control group. The major components of the intervention were: Sensitization of hospital management to prioritize PMTCT. The technical advisory group that was formed from the Sudan National AIDS program (SNAP), the representative of obstetrics and genecology specialists, and the senior management in hospitals all supported this. It was agreed that the HIV test will be added to the lab prescription that has ANC routine tests. They ensured that at least twice a week a quick orientation on PMTCT is done to the doctors in the hospital. This was included as a discussion point in their monthly meetings. Ensuring HIV testing kits are available and no stock-out during the intervention period. This was done in collaboration with SNAP, who confirmed supplies will be provided and no stock out will happen during the intervention period. The research team followed up on a monthly basis with the PMTCT team to ensure supplies are pre-positioned. Orientation of doctors with reminders to offer HIV tests during ANC visits. The senior genecologists in maternity hospitals were asked to facilitate this at least twice a week. The technical advisory group was assigned to monitor this. Incentivizing counsellors to increase group and individual awareness and counselling for pregnant women during waiting time as well as individual counselling. The research team provided technical and supportive supervision to the counsellors in the hospitals. A limited financial incentive was also secured for the counsellors. Work with the lab technicians to expect an increase in HIV test requests. The research team followed up with the lab teams in the hospitals who also received orientation from the hospital management on the expected increase in the number of HIV tests. Mother-to-mother peer support group. We considered the educational level of pregnant women and their exposure to PMTCT and created effective awareness and counselling sessions. A group of pregnant women volunteered to support PMTCT awareness. They were trained on interpersonal communication skills and developed awareness schedules targeting their peers during the ANC visits. Supported aid materials. Print and audio-visual Information, Education, and Communication (IEC) materials are available and clearly visible in the waiting areas. A set of IEC materials that address the knowledge gap in relation to the individual level determinants was drafted and pre-tested among the pregnant women. These materials were then produced and placed in the waiting areas of the PMTCT sites and small leaflets were also handed to pregnant women who joins the group awareness and counselling sessions. Reporting: The research team worked with the PMTCT teams in the hospitals to ensure that the routine reporting systems are strengthened to capture the intervention components. For example, the number of counselling sessions conducted, the number of pregnant women that participated, and the number of pregnant women tested. As described in the evaluation section that follows, these were captured before, during, and after the intervention. 

### 3.4. Step 6: Evaluation

The short-term impact in terms of increasing the utilization rates of the PMTCT services in selected hospitals is key in our evaluation plan comparing experimental and control groups [23]. To support this step of evaluation, pre- and post-assessments were carried out in all four hospitals. Data from pregnant women were collected from all four hospitals before, during, and after the intervention. In addition, hospital records on HIV test status during pregnancy were compiled before, during, and after the intervention. This data will be analysed in a time-series design to see the effect of the intervention, especially in the experimental hospitals. The self-reported data on HIV testing status will be compared with the actual administrative data from the hospital. Although the sample frame is different, we still believe that having these percentages will be useful to capture and present. Some of the limitations in the evaluation are related to the fact that this is not a randomized controlled trial (RCT), and pregnant women might be subjected to other interventions outside our plan, which make it challenging to attribute any changes in their behaviour exclusively to our intervention. Another factor is that the cohort of the pregnant women we interviewed before the intervention might not be the same as those who participated in the post-assessment after the intervention. We tried to mitigate this by giving priority to those who are in their first trimester as there are high chances of seeing them again after the period of six-month intervention. In addition, collecting the hospital data and analysing them before, during, and after our intervention will also provide a good indication that the findings can be (partially) attributed to our interventions. 

## 4. Discussion

This article outlines the development of an intervention aimed at increasing PMTCT uptake. It addresses behavioural and environmental changes to stimulate PMTCT uptake among pregnant women in Sudan. PMTCT services are important to eliminate the risk of MTCT; thereby, having zero new HIV infections among children. PTMCT interventions are important for pregnant women to make an informed decision regarding labour (caesarean section), breastfeeding, and childcare [5,6]. Our intervention plan was based on key factors from the findings of our pre-intervention assessment. These are the reluctance of medical doctors to offer HIV tests, pregnant women exhibiting fear and tension associated with HIV/AIDS, limited self-efficacy, and questioning confidentiality [15,16]. However, these also included other layers such as creating an enabling environment for doctors, improving counsellors’ work, and ensuring confidentiality, hospital management approval, and hospital policy directives. We considered all of these in developing our intervention. Our intervention plan has eight components designed to address the identified individual level determinants as well as the environmental ones. These components were derived from our own assessment and also supported by the international literature [13,15,16,20].

Key challenges we faced were the difficulty in brining various actors together, the reluctance of healthcare providers to offer HIV tests, and the low motivation of the hospital cadres. However, the management teams in the four hospitals were collaborative and supported the implementation of the intervention. 

The results of this study provide good insights into how future facility-based health promotion interventions that increase the utilization of PMTCT services can be developed and implemented. An evaluation of this intervention should be documented and presented to the policy and management teams in these hospitals and Sudan National AIDS control program (SNAP) for strengthening and scaling up PMTCT programs.

## 5. Conclusions

This paper provides insights into how to develop, implement, and evaluate a facility-based health promotion intervention. The pre-assessment was critical in shaping the intervention and making it relevant and evidence based. The Intervention Mapping approach that was applied facilitated the systematic design of the intervention and supported guiding the implementation. 

## Figures and Tables

**Figure 1 behavsci-13-00317-f001:**
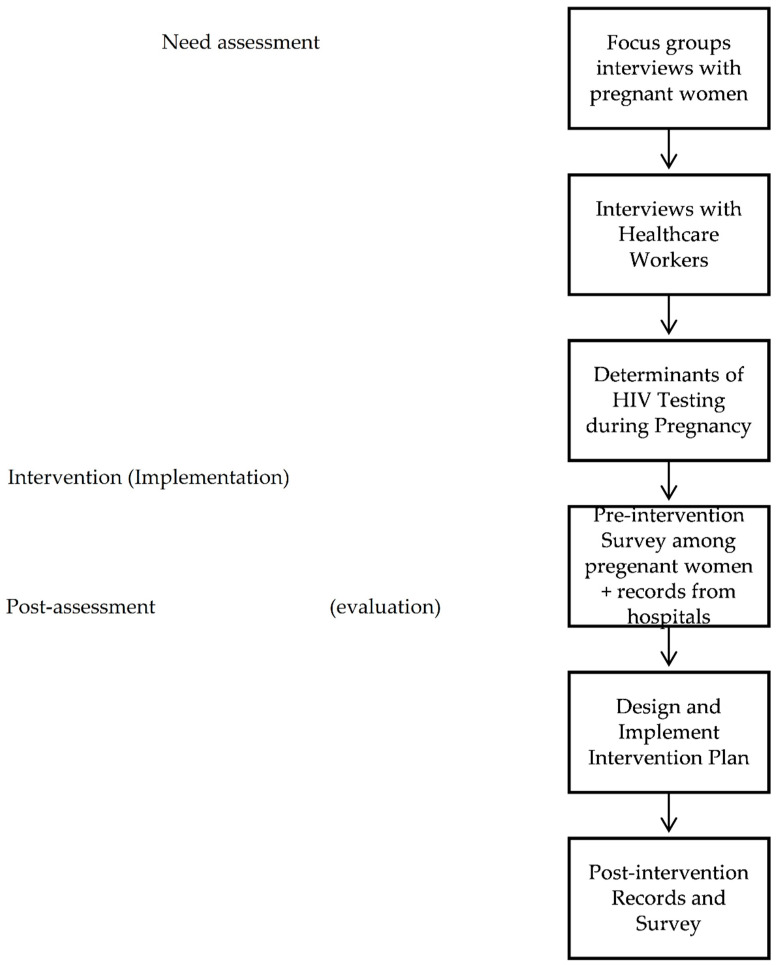
The intervention’s development, implementation, and evaluation, adapted from Intervention Mapping (IM).

**Table 1 behavsci-13-00317-t001:** Experimental and control hospitals.

Name of Hospital	Sample Size (Number of Pregnant Women)	Type of Group
Omdurman Maternity Hospital	126	Experimental (intervention)
Al Suadi Maternity Hospital	115	Control (no intervention)
Bahri Teaching Hospital	77	Control (no intervention)
Al Turki Naternity Hospital	67	Experimental (intervention)
Total	385	

**Table 2 behavsci-13-00317-t002:** Performance objectives and change objectives in environmental factors.

Existing Practices/Behaviours According to the Assessment	Determinants	Performance Objective	Proposed Interventions	Expected Effect
Strong coordination and advocacy with hospitals to support the intervention plan is needed	Willingness and support of the senior management in hospitals	Establish a technical advisory group to strengthen the coordination and ensure buy-in	Regular meetings, sensitization	Improved coordination and accountability to support the PMTCT intervention
Doctors are not offering the HIV test for the pregnant women	No clear policy directive. Very busy schedule and a lot of pregnant women coming, time-consuming Not a priority compared to other aspects of health care	Doctors offer HIV tests to pregnant women during their ANC visits	1. Sensitization of doctors, including the issuance of a clear policy directive from the director of the hospital 2. Key reminders and follow up with doctors	Increase in percent of pregnant women tested for HIV
Lab technicians/staff are not prepared for an additional number of HIV tests	Perceived as an additional burden	Ensure that testing kits for HIV tests are available Lab technicians are aware and committed to the expected increase in HIV tests	1. Provision of adequate testing kits 2. Reminders for the lab technicians	Quick and fast HIV tests are performed for pregnant women
Counsellors are not used to additional counselling sessions (pre and post-test sessions)	Perceived as an additional burden	Counsellors are prepared for additional counselling sessions following the expected increase of number of women who will be tested	1. Provision of supportive aid, e.g., IEC materials 2. Reminders and financial incentives for counsellors	Quality counselling sessions with more pregnant women involved

**Table 3 behavsci-13-00317-t003:** Performance objectives and change objectives in individual level determinants.

Existing Practices/Behaviours According to the Assessment	Determinants	Performance Objective	Proposed Interventions	Expected Effect
Pregnant women (PW) are not aware about the benefits of PMTCT	Limited knowledge about the PMTCT services and benefits Limited and poor public counselling sessions at PMTCT site	Ensure that 85% of pregnant women attending the ANC are exposed to HIV awareness Refresher training of the counsellors to ensure quality sessions	Awareness-raising sessions through counsellors and IEC materials (Print + TV screen) Mass-media short messages about importance of HIV test during pregnancy Peer education groups from PW themselves	Increase in knowledge among PW
Pregnant women are scared when they hear about HIV and AIDS	Scared of being positive and the consequences Believe that it is a chronic killing disease Lack of in-depth discussion on PMTCT	Ensure that 75% of pregnant women attending the ANC are exposed to quality HIV counselling	Counselling sessions targeting MTCT knowledge gaps Improved counselling by ensuring sessions are well-designed and tailored to the issues identified in our assessment Stigma reduction through giving examples of successful delivery of HIV positive women and the fact that with the ARV people with HIV lives productive life	Pregnant women are better prepared to accept HIV tests during pregnancy
Pregnant women are not convinced to undergo HIV test	Low self-efficacy Absence of high-quality counselling sessions	Ensure that 65% of pregnant women attending the ANC are exposed to HIV peer education and high-quality individual counselling sessions	Establishment of pregnant women peer groups to do peer education sessions	Increase in percent of women who accept HIV test

## Data Availability

The data presented in this study are available on request from the corresponding author.

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
