# Peer review of "Increasing Prevention of Mother to Child Transmission (PMTCT) Uptake through Facility-Based Health Promotion: Intervention Development"

_behavsci, 2023, doi:10.3390/bs13040317_

Round 1

Reviewer 1 Report

Dear Author,

This is a very interesting study, and I appreciate the difficulties in implementing this type of research. Below are my comments.

1. Advise including consenting participants in the Method section.

2. How were the women/patients participants incentivized for their participation?

3. How was it determined which hospitals would be control and intervention sites?

4. Ethical concern that control sites would never have access to the PMTCT intervention. Will this issue be addressed? 

5. No discussion about the next steps for the project? Will the authors pursue additional funding to expand this work? What are the prospects for the sustainability of the PMTCT programs in these hospitals? 

Reviewer 2 Report

Thank you for the manuscript on a priority health issue. We noted that most of the information and results compiled for developing the performance objectives and proposed interventions have been documented in two previous publications (ref. 15 and 16). However, it is unclear whether the current plan has been implemented. In fact, there are some elements of an experimental design (Table 1). If the aim is to develop and evaluate the impact of a behavior change intervention/strategy, other type of article should be written a with precise definition of context, study subjects, justified sample size, intervention procedures, outcome variables, timelines, etc. It is unclear whether this is to present/describe the building of a hospital-based approach, or to further test the imnpact of the specific model through a experimental design. The article should be more objective on its message, avoid duplication/fragmentation of information in the text, and explain where the PIs are in terms of the proposed plan (design phase, implementation or evaluation phase). Although this research topic is not innovative and may only apply to the local context, the theme is certainly of interest to those working in health behavioral sciences. Issues of feasibility, acceptability and measurement of impact need to be addressed.  Also, the conclusion section is not derived from the present article. In summary, the article needs to be revised before being considered for publication, the authors need to be more precise as to the purpose of this article, and how it relates with/complement the previous publications.

Round 2

Reviewer 2 Report

Thank you for the revised version of the manuscript, clarifying the critical points of the study.